# Vaccination of Elms against Dutch Elm Disease—Are the Associated Epiphytes and Endophytes Affected?

**DOI:** 10.3390/jof9030297

**Published:** 2023-02-24

**Authors:** Johanna Witzell, Caroline Sunnerstam, Tobias Hansson

**Affiliations:** Department of Forestry and Wood Technology, Linnaeus University, Georg Lückligs väg 1, 35195 Växjö, Sweden

**Keywords:** *Ulmus* spp., Dutch Trig^®^, vaccination, biocontrol, epiphytes, endophytes

## Abstract

Dutch elm disease (DED) is causing extensive mortality of ecologically and culturally valuable elm trees (*Ulmus* spp.). Treatment of elms with the biological vaccine Dutch Trig^®^ has been found to provide effective protection against DED by stimulating the defensive mechanisms of the trees. We hypothesized that the same mechanisms could also affect non-target organisms associated with elms. We explored the possible effects of vaccination on epiphytes (mainly lichens) and fungal endophytes living in the bark and young xylem of treated elms. Epiphyte cover percentage was assessed visually using a grid placed on the trunks, and a culture-based approach was used to study endophytes. Epiphyte cover was lower on the trunks of vaccinated trees as compared with unvaccinated trees, but the difference was not statistically significant. The presence of slow-growing and uncommon endophytes seemed to be reduced in continuously vaccinated elms; however, the highest endophyte diversity was found in elms four years after cessation of the vaccination treatments. Our findings suggest that although vaccination may shape epiphyte and endophyte communities in elms, its impacts are not straightforward. More detailed studies are, therefore, needed to inform the sustainable application of the vaccine as a part of the integrated management of DED.

## 1. Introduction

Elms (*Ulmus* spp.) are deciduous trees with a distribution area concentrated in temperate regions of the Northern Hemisphere [1]. In North America, six elm species are found, including the iconic American elm (*U. americana*) [2,3]. In Europe, three elm species (*U. glabra*, *U. minor* and *U. laevis*) are native, occurring especially near rivers and on floodplains, and provide multiple ecosystem services as components of mixed broadleaved forests. Elms provide valuable substrate and habitat for a rich biodiversity of insects, lichens, bryophytes and fungi, including several red-listed species that depend on them [4]. They are also popular urban trees, valued for their adaptability to a broad range of soil types and a high tolerance to adverse conditions, as well as their aesthetic habitus and canopy structure that make them ideal street trees [5].

Unfortunately, elms are highly vulnerable to a vascular wilt disease, Dutch elm disease (DED), which has spread in Europe and North America through two pandemics [6,7,8]. The first started in the early 20th century and was caused by a pathogenic fungus, *Ophiostoma ulmi* (Buisman) Nannfeldt. A more aggressive species, *O. novo-ulmi* Brasier has been spreading since the 1970s and is the causal agent of the second (current) pandemic. The disease develops when the pathogen spreads in the xylem, blocking it and thus causing the development of wilting symptoms [9]. Bark beetles act as vectors for the disease, with *Scolytus* or *Hylurgopinus* species being identified as the main genera responsible. The beetles lay their eggs under the bark of infected, dying trees. In spring, the adult beetles emerge and fly to new trees for maturation feeding, carrying fungal spores in their bodies and transmitting them to healthy trees [10,11]. The fungus is also known to be transmitted through root contacts between infected and healthy trees. Both the long-distance spread and the development of DED epidemics are associated with human activities. Trade and transport of infested elm logs or firewood across and between regions and continents has been a pathway for the spread of the disease to new locations. The extensive planting of elms as urban and amenity trees has facilitated the large-scale spread of the disease in the landscape, leading to the death of uncounted millions of elms [12].

Common measures to control DED include sanitation by pruning the symptomatic branches (initial stages), root graft severance between infected and healthy trees, and eradicating the infected trees before the insect flight season starts [13,14]. Breeding for resistance is a long-term strategy that has successfully produced clones with the capacity to cope well with the disease, providing a promising avenue for the restoration of elm stands in natural habitats [15,16]. In urban environments, a biological preventive treatment has been used successfully. The treatment is applied in the form of stem injections of a commercial product (Dutch Trig^®^) that contains conidiospores of the fungus *Verticillium albo-atrum* strain WCS850, an isolate that was obtained from a potato field in Flevoland, the Netherlands [13,14]. The commercial product was developed in 1992 and is currently used in several countries (The Netherlands, Germany, Canada, USA and Sweden) as a preventive treatment. The injections are given to healthy trees using a gouge pistol that pushes a small chisel through the bark and releases a sufficient dose of product to the latest annual rings. To reach new annual rings, the treatment must be repeated annually.

Vaccination of elms with Dutch Trig^®^ is reported to stimulate the trees’ defensive capacity; whether there is direct interaction between the DED fungus and the injected *Verticillium* fungus is not known [14]. Detailed information about the exact mechanisms that are induced by the vaccine is scarce, but localization of the pathogen near its entry point has been suggested to be an important resistance trait, involving rapidly inducible production of antifungal and occlusive chemical compounds (phenolic compounds, pectin, suberin, lignin, and mansonones) [17,18,19,20]. Many of these mechanisms are non-specific and thus could also be functional against other organisms that use elms as a habitat.

Like other trees, elms host a rich microbial biodiversity, including internal endophytic fungi that may protect their hosts against herbivores, pathogens or abiotic stressors such as drought [21,22]. These fungi spread to trees horizontally from the environment [23], and thus, they need to overcome the physical barriers and down-regulate the host’s defensive mechanisms to colonize the internal tissues successfully. In an earlier study, Martín et al. [24] compared the frequency and diversity of endophytic fungi and defensive phenolic metabolites in elm trees with genotypes known to differ in resistance to Dutch elm disease. They found that resistant *U. minor* and *U. pumila* genotypes hosted a lower frequency and diversity of fungal endophytes in the xylem than susceptible *U. minor* genotypes and suggested that the resistance mechanisms that are active against the pathogen could also suppress some non-pathogenic endophytic fungi. Thus, the defensive mechanisms induced by vaccination could potentially lead to changes in the endophytic microbiome and possibly also affect other organisms intimately associated with elms.

Here, we report results from two independent pilot studies conducted in two areas of Sweden where vaccination of elms has been a part of the management strategies. Our aim was to gather information about the potential consequences of vaccination treatment on two fractions of biodiversity associated with the bark of elms: epiphytic lichens and mosses (Study 1) and endophytic fungi in the bark and young xylem (Study 2). We hypothesized that the vaccination-induced changes in the defensive metabolism of elms may cause shifts in the epiphyte and fungal endophyte communities of elms.

## 2. Materials and Methods

Two independent studies on vaccinated vs. unvaccinated elms were conducted during March–May 2022, one focusing on the epiphytic flora in *U. glabra* var. *campertownii* in the Stockholm area and another on endophytic fungi on the island of Gotland (*U. minor*, *U. glabra*). 

### 2.1. Study 1—Epiphytes (Stockholm)

Study I was conducted in the cemetery area Norra begravningsplatsen in Solna (59°21′31.003″ N, 18°1′15.327″ E), north of Stockholm, where about 240 elm trees grow and where vaccination treatment has been a part of the protective management strategy since 2011. Seventeen inoculated and an equal number of uninoculated *U. glabra* ‘Camperdownii’ trees (approximately 100 years old) were selected for this study. For each of the trees, the percentage cover of epiphytes (mainly lichens) on the trunk was estimated from the four cardinal directions. This was performed by placing a grid consisting of 100 squares, each 2 cm × 2 cm, on the trunk at about 1.3 m above the ground. The number of squares in which epiphytic growth could be visually observed as presence/absence was recorded and expressed as a percentage of 100 squares (Figure 1). Thus, if epiphytic growth was observed in all 100 squares, the cover percentage was 100. The method did not provide information about the patchiness of the epiphytic coverage, nor did it separate between older and newer growth. The diameter at breast height (DBH, 1.3 m from ground level) of each tree was also measured because this has been found to correlate with characteristics, such as bark structure, habitat complexity and tree architecture, that can be significant for the epiphytes [25].

### 2.2. Study 2—Endophytes (Gotland)

For the study of endophytes, we collected samples from asymptomatic elms at three sites in southern Gotland in mid-March 2022. The sampled trees on sites II and III belonged to species *U. minor*, which is the dominant elm species in Gotland. At site I, however, some *U. glabra* trees had also been vaccinated. Because of the scarcity of suitable trees available for the study, these trees were also included. At site I, elms had been annually vaccinated since 2017 (“continuous vaccination”), whereas at site II, the elms had been vaccinated between 2014 and 2018 (“ceased vaccination”) as a part of the measures implemented in the European Union financed project “Saving wooded Natura 2000 habitats from invasive alien fungi species on the Island of Gotland, Sweden” (Reference: LIFE12 NAT/SE/001139, ELMIAS, 2013–2018). At site III, elms had not been vaccinated and are referred to as “unvaccinated” in this study. The three sites were located on private properties, where the elms were growing in cultural settings as alley or garden trees or in small groves. The distance between site II (56°58′10″ N, 18°13′43″ E) and the other sites (Site I: 57°12′14″ N, 18°21′22″ E; Site III: 57°12′24″ N 18°21′43″ E) was about 30 km. The elm trees selected for the sampling were mature large trees (exact age not known). To control the potential dependency between endophyte abundance and diversity and tree size, the DBH of each tree was measured. 

For the isolation of endophytes, samples were collected from 15 elms at each site (a total of 45 elms). To cause only minimal damage to the trees, we used a manual cork borer (diameter 8 mm) to cut plugs containing bark and young xylem (the borer reached 1–2 mm into the wood) at about 130 cm height on the east side of each tree. The samples (1 per tree) were placed individually under sterile water and transported to the laboratory. The cork borer was sterilized between the samples using 70% ethanol to avoid cross-contamination of the samples and to prevent transfer of fungi between the trees.

The samples were transported to the laboratory in cool boxes (ca. +7 °C). Within 24–48 h, the samples were surface sterilized by immersing them in the following series of solutions: NaOCl (sodium hypochlorite; bleach, ca. 2% solution, 45 s), sterile H_2_O (60 s), 70% ethanol (30 s) and a final rinsing in sterile H_2_O. Each plug was then split into four segments that were placed on the surface of Petri dishes (90 mm Ø) containing 1.5% water agar. Thus, a total of 180 segments were plated on as many water agar plates. Each plate was sealed with parafilm to prevent desiccation and stored in boxes that were kept at 20 °C in darkness. After 21 days, the plates were checked and the number of segments on each plate yielding hyphal growth was recorded to obtain an estimate of the frequency of fungal infections in each sample. 

Using heat-sterilized scalpels, small pieces of each homogenous hyphal aggregation that had emerged from the segments were sub-cultured once to fresh malt extract agar plates (90 mm Ø). Of the 252 sub-cultures to malt extract agar, 250 developed a growing colony within 14 days. After about 21 days, the cultures were grouped into 121 morphotaxa (Appendix A), except for 11 specimens that did not have clear mycelial characteristics and were excluded from further analysis. The main morphological traits used were colony color, surface structure and topography, the shape of the colony margins and pigmentation of the agar. Based on the growth rate, the isolates were also classified as exhibiting slow growth (colony diameter after 21 days less than 20 mm), intermediate growth (colony diameter about 20–40 mm) or fast growth (colony diameter > 40 mm per 21 days).

### 2.3. Data Analysis

The differences in epiphyte cover between vaccinated and unvaccinated trees were examined using Wilcoxon’s two-sample test. The difference in diameter between the unvaccinated and vaccinated trees was compared using a *t*-test and the relationship between tree diameter and epiphyte cover was examined using Spearman’s rank correlation.

The total number of isolates (N) and morphotype richness (S) of endophytes were recorded per site. Colonization frequency (CF) was calculated as the percentage of samples yielding at least one fungal isolate. The infection rate (IR; a proxy of fungal biomass) was determined as the average number of endophyte isolates recovered from each sample. There were no differences in CF but the differences in IR and the relative abundance of morphotaxa between the sites were tested using one-way ANOVA followed by a Tukey–Kramer HSD post hoc test. The diversity of endophyte morphotaxa at each site was evaluated using the Shannon diversity index [26]:(1)$$H^{\prime }=-\overset{s}{\underset{i=1}{\sum }}pi\ln pi$$
where *H*′ is the Shannon diversity index, *S* is the number of species (morphotaxa) and p*i* is the proportion of the specimens of the *i*-th species in the community. Differences in Shannon indices between the sites were evaluated using Hutcheson’s *t*-test [27]. Shannon equitability (evenness, E) was determined as *H*′/ln(*S*). To quantify how different the sites were in terms of the morphotaxa found at those sites, Bray–Curtis dissimilarity was calculated using the formula:(2)$$BC_{ij}=1-\frac{\left(2\times C_{ij}\right)}{\left(S_{i}+S_{j}\right)}$$
where *C_ij_* is the sum of the lesser values for each morphotaxa found in samples from each site, *S_i_* is the total number of specimens counted at site *i* and *S_j_* is the total number of specimens counted at site *j*. The median test was used to evaluate the difference in the relative abundances of morphotaxa between sites. Venn diagrams were plotted to display the number of shared morphotaxa vs. the number of morphotaxa unique to the sites. The relationship between tree diameter and the number of morphotaxa per tree was tested using linear correlation. The differences were considered statistically significant at α = 0.05. The analyses were conducted using JMP^®^ Pro 17.0.0 (JMP Statistical Discovery LLC, Buckinghamshire, UK).

## 3. Results

### 3.1. Study 1—Epiphytes (Stockholm)

The area covered by epiphytes was lower on vaccinated than unvaccinated trees; however, the difference was not statistically significant for the data from any of the cardinal orientations (Wilcoxon test, *p* > 0.05; Figure 2), nor for their mean value (Appendix A).

No difference was found in the mean DBH between the unvaccinated (29.9 ± 2.04 cm) and vaccinated trees (31.7 ± 1.53 cm) (*t*-test, *t* = 0.5677, *p* = 0.7127). There was a significant negative correlation between DBH and mean epiphyte cover on vaccinated trees (Spearman ρ = −0.493, *p* = 0.044) but not unvaccinated trees (Spearman ρ = −0.350, *p* = 0.168). For vaccinated trees, DBH correlated significantly with data from E and S orientations (Spearman ρ = −0.591, *p* = 0.012 and ρ = −0.521, *p* = 0.031, respectively), nearly significantly with data from the W orientation (Spearman ρ = −0.465, *p* = 0.059) and not significantly with data collected from the N orientation (Spearman ρ = −0.083, *p* = 0.750).

### 3.2. Study 2—Endophytes (Gotland)

In total, the four segments of each sample all yielded hyphal growth in water agar, resulting in a CF of 100% for all three groups (Table 1). The samples collected from site II yielded the highest number of isolates, with an average IR value of 1.48 (Table 1).

Morphotaxa richness (S) was highest in the samples collected from trees growing at site II, where the vaccinations ceased after 2018 (Table 1). The prevalence of singleton morphotypes was high in these trees (Table 1). The Shannon index (H) values also indicated the highest diversity for the samples collected from trees at site II (Table 1). Based on colony morphology compared with the previously identified specimens in our lab, we preliminarily identified *Aureobasidium pullulans* (de Bary) Arnaud as the most abundant fungus in the site II community. The lowest values of morphotaxa richness (S) and occurrence of singleton morphotaxa were found in samples from the continuously vaccinated trees at site 1 (Table 1). Bray–Curtis dissimilarity values suggested the highest compositional similarity in morphotype assemblage between sites I and II (0.84), compared with sites II and III (0.77) or sites I and III (0.72). The high evenness index values indicate that the endophyte communities were stable at all sites (Table 1).

A total of 46 morphotaxa (38% of all morphotaxa) were represented in the material by at least two specimens, but the majority (75 morphotaxa or 62%) of the morphotaxa were singletons, i.e., present only as single occurrences in all the material (Appendix A). The median test indicated that the relative abundance of morphotypes differed among the sample groups (Figure 3). The isolates growing at intermediate and fast growth rates dominated among the isolates recovered from the site I, whereas the slow and fast-growing ones dominated among isolates from trees at sites II and III (Table 1).

Six morphotaxa were recovered from all three sites (Figure 4). Samples from site II yielded the highest number of unique morphotaxa (only found at one site), whereas the lowest number of unique morphotaxa was recovered from site I samples (Figure 4). Site II shared more morphotaxa with site III than with site I (Figure 4).

The diameter of trees from site I (81.1 ±12.09 cm) was higher than that of trees from sites II (42.5 ± 2.18 cm) and III (49.2 ± 1.65 cm); however, there was no correlation between the diameter and number of isolates obtained from trees (r^2^ = 0.0172, *p* = 0.3901).

## 4. Discussion

Induction of plant defenses in response to chemical or biological stimuli is a promising approach in integrated pest management [28]. Induction of defenses is also the reported mechanism behind the vaccination treatment of elms against DED using the commercially available product Dutch Trig^®^ [14]. Mounting the induced defense responses is known to often involve energetic and metabolic trade-offs between defense and growth or reproduction in plants [29]; however, in most cases, the possible unintentional consequences of the inducive treatments for the multitude of non-target organisms living in intimate association with the host plants have remained understudied. Our studies provide a snapshot into the potential impacts of the responses induced by vaccination treatments of elms on their associated epiphytes and endophytes.

Our results provide some, but not unequivocal, support for our hypothesis that vaccination against DED may have unintentional consequences for the diversity and abundance of epiphytes (lichens and mosses) and endophytic fungi in the bark of elms. In particular, the presence of slow-growing and uncommon (singleton) endophytes seemed to be limited in elms that had been continuously vaccinated up to the sampling date. This could have resulted from vaccination-induced stimulation of defensive mechanisms in elms, which may also restrict the occurrence of some endophytes in elms that have been selected for their genetic resistance to DED [24]. The trend towards lower epiphyte cover in vaccinated elms suggests that the vaccination-induced effects could be tangible even on the bark. Stemflow and canopy throughfall chemistry [30], as well as bark acidity [31], have been identified as important factors for epiphytic lichens, but it is not known if the induction of resistance mechanisms by vaccination could change these factors, e.g., through changes in leachate composition.

Intriguingly, and contrary to our hypothesis, the most diverse communities of the culturable endophytes seemed to be associated with elms that had been vaccinated during a four-year period but not since 2018 (i.e., four years before the sampling took place). We cannot exclude the possibility that the observed pattern was merely a reflection of the geographic distance between site II and the other two sites. The vaccination of elms on both sites I and II was conducted as a part of the practical management strategies and optimally comparable unvaccinated control trees within the sites were, unfortunately, not available. Therefore, the results from the different sites may be strongly affected by site-specific variations that governed the horizontal transmission of endophytes from the surrounding environment [23]. In our study, the fact that more slow-growing morphotaxa were recovered from site II samples than from site I and III samples could also reflect the local differences in the quality of the environmental inoculum. Moreover, the genetic background and historical conditions experienced by the trees at the sites could influence the endophytic assemblages [32,33,34]. On the other hand, the endophyte communities of trees from the closely located sites I and III were not identical in terms of morphotype diversity or abundance, and thus, neither can we exclude the possibility that a certain amount of selection occurred due to the vaccination treatment. In summary, our findings suggest that although vaccination treatment may shape the associated endophyte biodiversity, its effect is not straightforward temporally or spatially.

Taxonomic identification of epiphytes was not undertaken for the estimation of epiphyte cover in our pilot study, and therefore, detailed conclusions about the diversity in these communities cannot be drawn. However, in general, crustose and foliose lichens seemed to be the most common types of epiphytes observed on elms. The result that the epiphyte cover tended to be highest on the north-facing side of the tree bole indicates advantageous conditions for lichen growth on this side of the trees, possibly including a good balance between sunlight and moisture, which lichens require to avoid desiccation and to maintain photosynthesis [35,36]. This contrasts with the findings of Jüriado and co-workers [2], who studied epiphytic lichen species richness, cover and composition in deciduous trees including *U. glabra* and *U. laevis* in a floodplain forest in Estonia. They found that the cover of lichen species did not show a significant response to the cardinal orientation of the tree trunk, although the composition did [31]. The discrepancy between our result and theirs can probably be explained by differences between the study sites. In conditions where light exposure is higher, such as the open area in our study, the impact of abiotic factors (light exposure, temperature, variations in humidity, e.g., due to wind) can be more pronounced than in a forest [31,37]. We expected the epiphyte cover percentage to correlate positively with DBH, because the sizes and ages of trees have been found to explain variation in epiphytic species composition and diversity in trees [38]. Instead, the DBH significantly negatively correlated with epiphyte cover in vaccinated trees. The negative relationship may have been due to the saturation of suitable establishment sites for epiphytes on the studied elms [25] and the limited presence of dispersing lichen propagules in the surrounding urban environment. The stronger correlation in vaccinated trees is likely to reflect the generally lower epiphyte cover in these trees.

Only a fraction of fungal endophytes can be captured using the culture-based approach [39], but the morphotaxa-based characterization of endophytic communities does not reveal the detailed taxonomic diversity of this fraction [40]. However, the morphotaxa provide a starting point to explore the differences in endophyte diversity in vaccinated and unvaccinated elms. The high abundance of the preliminarily identified *A. pullulans* isolates from the site II samples suggests a role for this fungus in the post-vaccination endophyte community. In earlier studies, this ubiquitous fungus has been reported as a common inhabitant of elms [24,41,42], as well as in other woody plants such as *Populus* species [43,44,45] and grapevine (*Vitis vinifera*) [46]. *Aureobasidium pullulans* is also known for its biocontrol activity [47], which is possibly related to its capacity to produce a variety of antimicrobial toxins and enzymes [44]. Similar to species of *Ophiostoma*, *Aureobasidium* species are also known as blue stain fungi that discolor wood and wood protective coatings. Hypothetically, *A. pullulans* could compete with *O. novo-ulmi* for the same niche in elms. In an earlier study, Blumenstein et al. [48] investigated the nutritional niche overlap between selected isolates of *O. novo-ulmi* and *A. pullulans* and found that although the two species indeed shared a large part of a nutritional niche, *A. pullulans* was a weak competitor for substrates. Thus, the potential relevance of the high abundance of *A. pullulans* in elms after the vaccination treatments had been terminated remains unclear. The six endophyte morphotaxa found at all three sites may belong to a core endophyte microbiome [49,50] of elms, representing fungal taxa that are consistently resident in elms regardless of the habitat. The shared taxa are assumed to represent the most ecologically and functionally important microbes for the host [51] (Neu et al., 2021). The fact that the highest relative abundances of these morphotaxa were found in vaccinated trees could reflect the importance of these endophytes for the defensive capacity of elms.

## 5. Conclusions

According to the concept of extended phenotype, epiphytes and endophytes can influence the host plant phenotype by expanding its genomic and metabolic capabilities and thus influencing its fitness [52]. The recently emerged view that plants should be considered as holobionts, i.e., functional assemblages of plant and microbial species [53,54], emphasizes the importance of understanding the dynamics of tree-associated biodiversity. Epiphytes and endophytic microbes have significant ecological functions, and their biotechnological potential is still largely unexplored [55,56,57,58]. Therefore, and also considering the global concern for biodiversity loss (UN Sustainable Development Goal, SDG 15), it is important to ensure that biopesticides such as Dutch Trig^®^ do not have direct or indirect adverse effects on the associated biodiversity.

Despite the limitations of our approach, the results suggest the value of continued studies on the topic. These should be carried out using advanced molecular methods that allow a detailed description of the taxonomic and functional diversity of epiphytes and endophytes. Controlled experiments are necessary to thoroughly assess the potential impacts of vaccination on the biodiversity associated with elms. We emphasize that exploring this topic does not argue against the general usefulness of the vaccination treatment as a measure for protecting healthy elms against DED, especially in urban settings. Rather, detailed information about the effects of the treatment on the associated biodiversity could support the design of guidelines to optimize the sustainability of the treatment. For instance, measures such as transplanting of bark-associated lichens [59] could be applied where support for epiphyte diversity is deemed necessary. More detailed studies on the impacts of vaccination-induced effects in elms may also lead to an increased understanding of the different components of elm resistance against DED.

## Figures and Tables

**Figure 1 jof-09-00297-f001:**
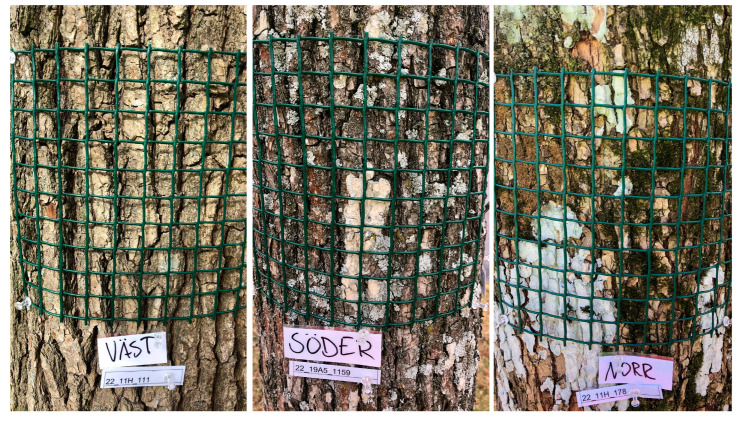
In the epiphyte cover survey, a sampling grid (20 cm × 20 cm) divided into 100 squares was positioned in the four cardinal orientations (N, E, S, and W) on each tree. The frequency (occurrence within squares) of epiphytic growth was determined. Photo: Tobias Hansson.

**Figure 2 jof-09-00297-f002:**
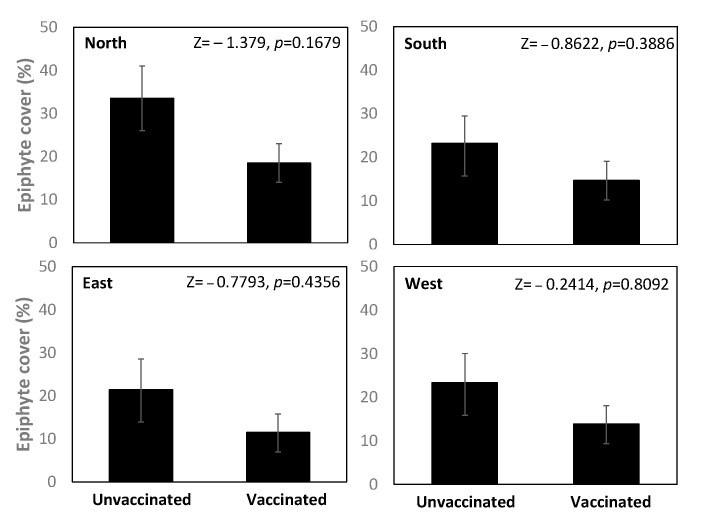
Epiphyte cover (%) on the trunks of unvaccinated and vaccinated elms was surveyed in the four cardinal directions (N, S, E, W). Shown are the mean values of 17 replicates and Wilcoxon two-sample test results. Vertical bars represent the standard error of the mean (*n* = 17).

**Figure 3 jof-09-00297-f003:**
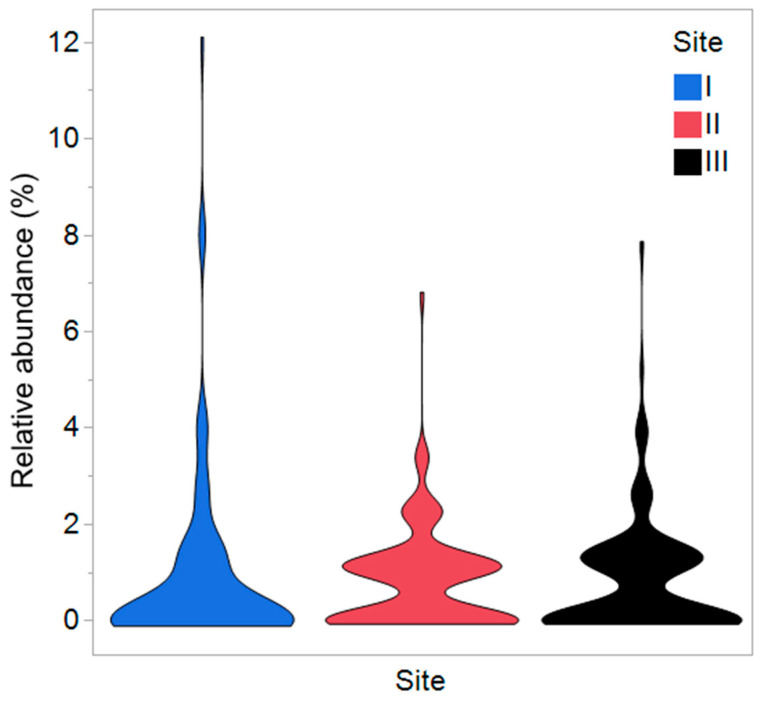
Violin plot of relative abundance (%) of 121 morphotaxa in bark and xylem samples collected from elms growing at the three sites where vaccinations were ongoing at the time of sampling (Site I), where vaccinations had ceased (Site II) and where trees had not been vaccinated (Site III). Median test: ChiSquare 13.58, df = 2, *p* = 0.0011.

**Figure 4 jof-09-00297-f004:**
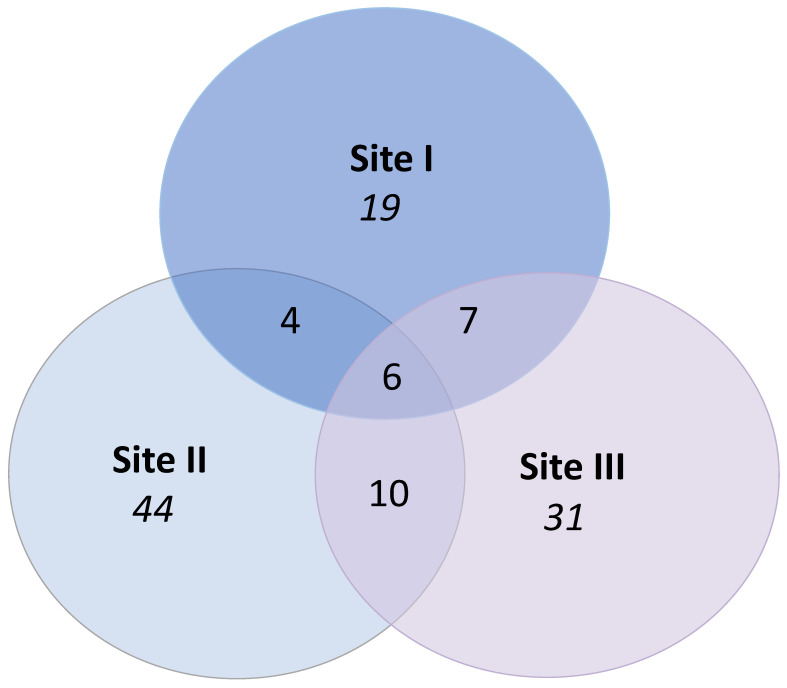
Venn diagram displaying the number of fungal endophyte morphotaxa detected in the samples from elm trees growing at the three sites where vaccinations were ongoing at the time of sampling (Site I), where vaccinations had ceased (Site II) and where trees had not been vaccinated (Site III). Numbers in the overlapping regions indicate shared morphotaxa between the groups. Numbers in the non-overlapping regions indicate unique morphotaxa for the group.

**Table 1 jof-09-00297-t001:** Summary of the results for endophyte frequency and diversity, based on analysis of bark and young xylem samples collected from elm trees growing at the three sites where vaccinations were ongoing at the time of sampling (Site I), where vaccinations had ceased (Site II) and where trees had not been vaccinated (Site III). Mean values with different letters are significantly different (α = 0.05). EF = endophyte, MT = morphotaxa.

	Site I	Site II	Site III
** *Endophyte frequency* **			
Nr of segments plated	60	60	60
Nr of segments yielding EF	60	60	10
Colonization frequency (CF, %)	100	100	100
Total nr of EF isolates (N)	75	89	77
Isolation rate (IR)	1.25 a	1.48 b	1.28 ab
Nr of isolates belonging to “core community” morphotypes	26	10	15
** *Endophyte diversity* **			
Nr of MTs (Richness, S)	36	64	54
* Nr of common * MTs*	*24*	*29*	*26*
* Nr of singleton ** MTs*	*12*	*35*	*28*
Shannon diversity index (*H*)	3.32 a	4.01 b	3.85 a
Shannon equitability (*H*/ln*S*)	0.92	0.96	0.96
** *Growth rate of colonies* **			
Slow	13	57	34
Medium	36	12	12
Fast	51	31	54

* Present in the material as at least two isolates. ** Present in the material as only one isolate.

## Data Availability

The raw data are made available in Zenodo (10.5281/zenodo.7558609) when the manuscript is published.

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
