# Peer review of "Vaccination of Elms against Dutch Elm Disease—Are the Associated Epiphytes and Endophytes Affected?"

_jof, 2023, doi:10.3390/jof9030297_

Round 1
Reviewer 1 Report
The study by Witzell et al. investigates the potential effects of applying a commercial anti-DED fungal product on parts of the holobiont of injected elm trees. The research hypothesis thus was the potential reduction of associated endophytes and epiphytes through vaccination of elm trees. Reasons could be induced defensive reactions.
My specific comments follow here:
L60-61: can you provide a bit more details on the development of the Dutch trig product?
L96-98: It is not clear why not both endophytes and epiphytes were examined on both locations. Please provide and explanation here.
L99-100: please add that both U. glabra as well as U. minor were studied on Gotland.
L106-107: please replace with: Seventeen inoculated and an equal number of uninoculated U. glabra 'Camperdownii' trees (approximately 100 years old) were selected for this study.
L110-112: question: was the % of square area covered by epiphytic growth also taken into account, or only presence/absence of any growth? Or was a certain minimum % cover considered within a square to assign yes/no for epiphytic growth? Patchiness versus continuous cover? Were differences considered between older and newer growth?
L130: please put U. minor in italics.
L154: NaOCl (sodium hypochlorite; bleach).
L157: can you provide more details about the incubation conditions here? (humidity, agitation or not, incubator model)
L162: these were subcultured only once, or multiple times?
Legend Figure 2: there is a space missing between N, S
L228: ...at least one colony to water agar? Please re-phrase.
Table 1: there seems to be a formatting problem with the second column (the superscript a is hardly readable). Also, please be consistent with calling Sites I, II, III (column names).
Figure 3 shows that some morphotaxa have very high relative abundance on site I (ongoing vaccination). Can you describe them here? Related to information provided in L368-371?
L344-345: please review reference format.
L373-375: there is also the concept of the 'extended phenotype' that can be introduced here.
It would have been interesting to have measured some known (highly effective) defense compounds in the screened elm trees, to see if there is a correlation between certain defense compounds (qualitatively and quantitatively) and the observed patterns in epiphyte and endophyte occurences.
Greenhouse experiments could have been added to control for site-specific variation in endophytes composition.
Author Response
We thank the Reviewer for constructive comments and questions that were much apprciated. Please find the attached file with detailed responses. The revisions are marked with yellow in the manuscript which has also been edited by a professional language editor (Sees-Editing Ltd., UK). We hope that the manuscript has now been improved to allow publishing.
On behalf of authors,
Johanna Witzell

Reviewer 2 Report
I am not familiar with studies that group fungi into “morphotypes” (based on growth rates) – this provides little information. I was expecting some idea of what species or Genera were identified and how they varied across the treatments. There are many modern tools that allow for species designations (molecular markers) or at least to the level of Genera. Ironically, the authors refer to citation 50 – which uses amplicon sequencing to at least define OTUs. I agree this study provides some information on the effects of Dutch Drig®. In summary, this study has its merits and the hypothesis to be tested is well formulated. BUT without species identification I find this study of limited interest. The paper could benefit from editing and should be streamlined (shortened).
Line 40: 1970s not 1970ies
Line 41: Ophiostoma-fungus? Avoid this term - you should be able to be specific with regards to what Ophiostoma species is/are present.
Line 325-326 – What robust method are the authors referring to?
Line 358: Ophiostoma-species no hyphen needed; “Similar to species of Ophiostoma ……. “
Author Response
We have now considered all the comments and suggestions and addressed each of them. A detailed list of our responses is presented in the attachment.
We thank the Referee for the constructive and helpful comments and hope that our revision has adequately improved the manuscript.
The language has been edited by Sees-Editing Ldt, UK.
On behalf of authors,
Johanna Witzell

Round 2
Reviewer 2 Report
The authors have responded to my comments and made some changes, they did address my concerns with regards to the limitations of their study. Overall there are some interesting findings and that warrant publication.